# Design and Application of Electrochemical Sensors with Metal–Organic Frameworks as the Electrode Materials or Signal Tags

**DOI:** 10.3390/nano12183248

**Published:** 2022-09-19

**Authors:** Yong Chang, Jiaxin Lou, Luyao Yang, Miaomiao Liu, Ning Xia, Lin Liu

**Affiliations:** 1College of Chemistry and Chemical Engineering, Anyang Normal University, Anyang 455000, China; 2School of Chemistry and Materials Engineering, Jiangnan University, Wuxi 214122, China

**Keywords:** metal–organic frameworks, electrochemical sensors, small molecules, biomarkers, signal amplification

## Abstract

Metal–organic frameworks (MOFs) with fascinating chemical and physical properties have attracted immense interest from researchers regarding the construction of electrochemical sensors. In this work, we review the most recent advancements of MOF−based electrochemical sensors for the detection of electroactive small molecules and biological macromolecules (e.g., DNA, proteins, and enzymes). The types and functions of MOF−based nanomaterials in terms of the design of electrochemical sensors are also discussed. Furthermore, the limitations and challenges of MOF−based electrochemical sensing devices are explored. This work should be invaluable for the development of MOF−based advanced sensing platforms.

## 1. Introduction

More and more researchers are now committed to developing various novel methods for the detection of inorganic ions, organic pollutants, and biomarkers with the help of fluorescence, colorimetry, surface-enhanced Raman spectroscopy, electrochemistry, electrochemiluminescence, and photoelectrochemistry [1,2,3]. In contrast to the traditional analytical techniques of mass spectrometry and high-performance liquid chromatography, these methods offer the inherent merits of simple operation and high sensitivity. Among them, electrochemical sensors are extensively utilized in the fields of disease diagnosis, food safety, and environmental science due to their advantages of high sensitivity and selectivity, rapid response, low investment, and good portability. 

Electrochemical sensing performance can be improved by modifying the electrode surface and integrating various signal-amplified strategies, such as enzyme catalysis, DNA self-assembly, and nanomaterials-assisted amplification [4,5,6,7]. Because of their versatile electrical, mechanical, and physiochemical properties, nanomaterials, including graphene, quantum dots (QDs), and metal nanoparticles, have endowed electrochemical sensors with more possibilities in terms of sensitivity, selectivity, and real-time detection. In fact, the modification of an electrode with nanomaterials can not only increase the effective surface area, enhance the conductivity, and provide catalytic sites (e.g., oxidation metal sites and conjugated π-electron systems) but also facilitate the immobilization of biomolecules (e.g., enzymes, antibodies, DNA, and peptides) with the objective of improving the specificity and sensitivity. In addition, when acting as signal labels in bioassays, nanomaterials labeled with biorecognition elements and signal reporters can increase the number of reporters for each recognition event, finally leading to the amplification of the electrochemical signal. 

As a novel class of organic–inorganic hybrid crystalline porous coordination polymers, metal–organic frameworks (MOFs) that are self-assembled from metal ions/clusters and organic ligands exhibit many outstanding properties, including high tunable porosity, good stability, large surface area, and adjustable chemical functionalities [8]. A wide range of building blocks, such as metal ions and organic ligands, can endow MOFs with many effective functions for application in electrochemical assays (Figure 1). A wide variety of MOFs can be prepared using different approaches, including the solvothermal/hydrothermal method, sonochemical synthesis, electrochemical synthesis, microwave-assisted synthesis, and so on. The advantages of MOFs perfectly match the materials necessary for sensor fabrication [9]. For example, the large surface areas and porous nanostructures of MOFs offer more interfaces and accessible active metal sites by which to catalyze the electrochemical reaction of analysts on the electrode surface [10]. The abundant functional groups and mesoporous properties of MOFs facilitate their functionalization with various materials, including small molecules, antibodies, nucleic acids, enzymes, and nanoparticles. Therefore, MOFs have been used to construct electrochemical sensors with excellent performances in a broad range of potential applications [11,12,13].

The published MOF−based studies have been summarized in previous reviews [14,15,16,17,18]. For example, Li’s group and Jouyban’s group have reviewed the recent advancements of aptamer-functionalized MOF−based biosensors [19,20]. The preparation and application of MOF−based nanozymes in biosensing and cancer therapy have been reviewed by several groups [21,22,23]. MOFs can be used as luminescent and electrochemical probes for the detection of biological molecules and cancer biomarkers [24,25,26,27]. Wang et al. focused on MOF−based sensing platforms for virus detection [28]. Gorle et al. discussed recent achievements in MOFs−based composites for constructing electrochemical glucose sensors [29]. Wang et al. summarized the recent progress in sensing applications of metal nanoparticles/MOF composites [30]. Mohammadreza’s group reviewed the advancement of MOFs in the electrochemical sensing of environmental pollutants [31]. In recent years, MOF−based electrochemical sensors have aroused intense interest. However, there is little in the way of a systematic overview of MOF−based electrochemical sensors using MOFs as the electrode materials or signal tags [32]. Herein, we summarize recent progress in the design and application of MOF−based sensing platforms (Figure 2). Firstly, we briefly introduce the utilization of pristine MOFs and MOF−based composites as electrode materials for the electrochemical detection of electroactive species. Then, the design and application of MOF−based sensors were discussed, according to the functions of MOFs in the sensing devices. Generally, MOFs can be used as electrode substrates, nanocarriers for signal reporters, electroactive labels, electrocatalysts, and sacrificial templates. Finally, the limitations and challenges of MOF−based sensors are discussed, which is of great significance for exploring new multifunctional MOFs for electrochemical analysis. Due to the explosive growth of academic articles and the highly dynamic development of this topic, some important papers may be omitted during the review period. We sincerely apologize to those authors whose studies were overlooked in this article.

## 2. MOFs as the Electrode Materials of Electrochemical Sensors

MOFs, with unsaturated active sites, tunable pore sizes, and good electrocatalytic activity, can effectively enhance the electrochemical response and increase detection sensitivity. In addition, their high porosity, large surface area, and abundant functional groups are beneficial for concentrating the analytes and improving mass transfer efficiency [33,34,35]. The specific size and shape of the available cavities and channels can endow MOF−based sensors with relatively high selectivity. However, the inherent characteristics of low chemical stability in aqueous medium and the poor conductivity of MOFs may dramatically hinder their electrochemical applications in practical samples. Aiming to overcome the shortcomings, different conductive materials and nanomaterials have been integrated with pristine MOFs to improve the electrochemical sensing performance [36]. In this section, the advancements of MOFs in sensing electroactive small molecules and metal ions are briefly discussed, based on the types of additional materials that have been used for integration with MOFs.

### 2.1. Pristine MOFs

Inorganic nodes and organic linkers in MOFs play a decisive role in their electrocatalytic performances [37,38,39]. MOFs have been considered the ideal materials for non-enzymatic electrochemical sensors owing to the redox behavior of metal ions (Table 1) [40,41]. Several small biomolecules (e.g., dopamine (DA), uric acid (UA), ascorbic acid (AA), glucose, H_2_O_2_, and amino acids) have been catalytically oxidized by the active metal sites (e.g., Cu^2+^, Co^2+^, Cr^2+^, Zn^2+^, and Ni^2+^) in MOFs [42,43,44]. For example, Cu−based pristine MOFs could be used to detect small molecules due to the rapid electron transfer process of Cu(II)−MOF/Cu(I)−MOF couple. Moallem et al. prepared Cu-benzene-1,3,5-tricarboxylic acid (Cu−BTC) MOFs by the ultrasound-assisted hydrothermal method and used the composites to modify a carbon paste electrode (CPE) for the simultaneous detection of DA and UA [45]. Li et al. reported a Cr−MOF−based electrochemical sensor for the simultaneous determination of DA and UA [46]. The large pore volume of Cr−MOFs provided plenty of active sites to catalyze the oxidation of DA and UA, leading to an increased peak separation value. Co−MOFs with peroxidase-like activity have also been used as active substrates for the electrocatalytic reduction of H_2_O_2_ [47]. 

Environmental pollutants (e.g., pesticides, heavy metal ions, phenolics, and toxins) are harmful to both the environment and human health. MOF−based electrochemical sensors have been extensively developed for the analysis of environmental pollutants [48,49,50]. For example, Cu−TCPP and Cu−BTC were used as electrode modifiers to construct electrochemical sensors for the detection of glyphosate [51,52]. Hu et al. reported a Cu−MOFs−based sensor for the detection of carbendazim in water [53]. Dong et al. developed an electrochemical sensor for 2,4-dichlorophenol, based on Cu−BTC MOFs [54]. Heavy metal ions in natural resources are harmful to living organisms. Because of their characteristics of a large surface area, high porosity, and tunable chemical functionality, MOFs modified at the electrode can act as concentrators to accelerate the preconcentration process of anodic stripping voltammetry (ASV), improving the detection sensitivity [55,56]. For example, Guo et al. developed an electrochemical sensor for Pb^2+^ detection using amino-functionalized Ni−MOFs [57]. The fern-leaf-like MIL-47(as) was used to develop an electrochemical sensor for the simultaneous detection of Pb^2+^, Cu^2+^, and Hg^2+^ [58]. In addition, Guo et al. reported an electrochemical sensor for Pb^2+^ detection created by using amino-functionalized Ni(II)−based MOFs to modify the electrode [57]. Pb^2+^ was absorbed by the amino group of a 2-aminobenzenedicarboxylic acid (NH_2_−BDC) linker, leading to an increase in the current intensity. The Ni−MOFs−modified electrode showed excellent selectivity toward Pb^2+^, due to the ion size exclusion.

Compared with monometallic MOFs, bimetallic MOFs with an optimized ratio of metal ions exhibit higher electrical conductivity, stability, and catalytic efficiency [59,60]. Bimetallic MOFs (e.g., Cu−Co−MOFs, Zn−Ni−MOFs, and Co−Ni−MOFs) have been used to develop enzyme-free electrochemical sensors for the detection of various targets, including glucose, adenosine, organophosphate, bisphenol A (BPA), and so on [61,62,63,64]. Typically, Huang et al. developed a three-dimensional (3D) Co−doped Ni−based conductive MOFs−modified electrochemical sensor for the detection of L-tryptophan (Trp) [65]. Co–Ni−MOFs showed high catalytic activity toward Trp oxidation due to the porous structure, larger surface area, and more active sites in the redox process. The proposed sensor for Trp detection achieved a wide linear range from 0.01 to 300 μM and a low limit of detection (LOD) (8.7 nM). Compared to the conventional 3D MOFs, 2D MOFs are more attractive because of their outstanding advantages of larger surface area/volume ratio, easier diffusion, and more available active sites for catalysis. Li et al. fabricated an ultrathin Ni−MOF nanosheet-based assembly for the detection of AA (Figure 1A) [66]. The nanosheet, with a uniform thickness of 8 nm, facilitated the interaction between MOF and AA. The Ni−MOF nanosheet-modified electrode exhibited a satisfactory linear range from 0.5 μM to 8065.5 μM and an LOD of 0.25 μM. Wang et al. prepared a series of Co, Cu, and Zn-based ultrathin 2D bimetallic MOF nanosheets using Fe(III) tetra(4-carboxyphenyl)porphine chloride (TCPP(Fe)) as the ligand (Figure 1B) [67]. The MOF nanosheets obtained by the surfactant-assisted method formed a multilayer film on the electrode surface. The modified electrode was used for the sensitive detection of H_2_O_2_ secreted by live cells.

Porphyrins with stable and rigid structures, coordinated with different metal ions, can be used as electrocatalytic sites in a metal-binding or metal-free state [70]. Porphyrinic MOFs with excellent enzyme-mimicking catalytic activity have been utilized for the electrochemical detection of nitroaromatic compounds, nitrite, UA, H_2_O_2_, and so on [71,72,73,74,75]. For example, Kung et al. demonstrated that the zirconium-based porphyrin MOF (MOF−525) thin-film-modified electrode showed high conductivity and electrocatalytic activity for nitrite oxidation [76]. In view of the fast redox reaction and high catalytic activity of iron phthalocyanine (PcFe), Zeng et al. reported the detection of trichloroacetic acid (TCAA) using PcFe− and Zn−based MOF (ZIF-8) composites as the electrode materials (Figure 1C) [68]. In this method, PcFe(II) was first electrochemically reduced into PcFe(I), then the produced PcFe(I) was immediately re-oxidized by TCAA. Based on the redox cycling, TCAA has been determined with an LOD of 1.89 nM. Moreover, hexasubstituted triphenylene with good electrical conductivity has been used to prepare conductive MOFs. Ko et al. prepared an electrochemical sensor for the detection of redox-active neurochemicals using the 2D layered conductive MOFs−casted electrode, including AA, DA, UA, and serotonin (5-HT) (Figure 1D) [69]. Among the conductive MOFs, M_3_HXTP_2_ (M = Ni, Cu; and X = NH, 2,3,6,7,10,11-hexaiminotriphenylene (HITP) or O,2,3,6,7,10,11-hexahydroxytriphenylene (HHTP)), the Ni_3_HHTP_2_ MOFs exhibited the best sensing performances and could be used for the detection of 63 nM DA and 40 nM 5-HT, with a wide linear range. The sensor was further used for the determination of 5-HT in simulated urine.

**Table 1 nanomaterials-12-03248-t001:** Detection performances of pristine MOF−based electrochemical sensors.

Electrode Material	Analyte	Linear Range	LOD	Ref.
Zn_4_O(BDC)_3_ (MOF-5)	Pb^2+^	10 nM~1.0 μM	4.9 nM	[33]
TMU-16-NH_2_(Zn)	Cd^2+^	62.5 nM~1.1 μM	1.8 nM	[34]
[H_2_N(CH_3_)_2_]_4_[Zn_3_(Hdpa)_2_]•4DMF	Cu^2+^	5.0 pM~900 nM	1.0 pM	[35]
Co−MOFs	glucose	1.0 μM~3.0 mM	1.3 nM	[37]
Cu−MOFs	BPA	50 nM~3.0 μM	13 nM	[38]
Cu−BTC	methocarbamol	80 μM~800 μM	50 nM	[39]
Cr−MOFs	H_2_O_2_	25 μM~500 μM	3.52 μM	[40]
Cu−BTC	DA and UA	50 nM~500 μM and 0.5 μM~600 μM	30 and 200 nM	[45]
Cu_3_(BTC)_2_	2,4-dichlorophenol	40 nM~1.0 μM	9.0 nM	[54]
Co−MOFs	H_2_O_2_	5.0 μM~9.0 mM	3.76 μM	[47]
Ni−MOFs	Pb^2+^	0.5 μM~6.0 μM	0.508 μM	[57]
MIL−101(Cr)	DA and UA	5.0~250 μM and 30~200 μM	Not reported	[46]
Co–Ni−MOFs	Trp	10 nM~300 μM	8.7 nM	[65]
Ni−MOFs	AA	0.5 μM~8.1 μM	0.25 μM	[66]
(Co−TCPP(Fe))5	H_2_O_2_	0.4 μM~50 μM	0.15 μM	[67]
MOF−52(Zr)	NO_3_^−^	20 μM~800 μM	2.1 μM	[76]
PcFe@ZIF-8	TCAA	20 nM~1.0 μM	1.89 nM	[68]

**Abbreviations:** BDC, 1,4-dicarboxybenzene; MOFs, metal–organic framework; Hdpa, 3,4-di(3,5-dicarboxyphenyl)phthalic acid; BPA, bisphenol A; BTC, 1,3,5-benzenetricarboxylic acid; DA, dopamine; UA, uric acid; Trp, L-tryptophan; AA, ascorbic acid; TCPP(Fe), Fe(III) tetra(4-carboxyphenyl)porphine chloride); PcFe, iron(II) phthalocyanine; TCAA, trichloroacetic acid.

### 2.2. Carbon Materials-Modified MOFs

Owing to their high conductivity, low cost, and chemical inertness, distinct-dimension carbon materials have been extensively used to prepare hybrid composites with MOFs, including 1D carbon nanotubes (CNT), 2D graphene, and mesoporous carbon [77,78,79]. CNT is considered to be one of the most promising carbon materials to improve the performance of MOFs because of its unique electrical conductivity, high surface area, and high thermal/mechanical stability [80]. CNT/MOFs composites have been used for the electrochemical detection of a wide range of targets, including gallic acid, opioid drugs, tetracycline, and so on (Table 2) [81,82,83,84,85]. For instance, Li et al. reported an electrochemical sensor for the detection of DA and acetaminophen (ACOP), using the hybrid composite of UiO-66-NH_2_ and CNT as the electrocatalyst [86]. Mn−MOFs were grown in situ on multi-walled CNTs (MWCNTs) for the simultaneous detection of AA, DA, and UA in bodily fluids [87]. Moreover, quasi-2D Ni−MOF nanosheets were prepared in situ under the confinement effect of the cross-linked CNT networks; the resulting hybrid composites were applied to detect BPA [88]. Ratiometric electrochemical sensors, based on the ratio of an internal reference signal and a response signal for an analyte, can improve their accuracy and stability. Recently, Rong et al. designed a ratiometric electrochemical sensor for the determination of doxorubicin, based on electroactive methylene blue (MB)-loaded MWCNTs/UiO-66-NH_2_ [89]. As shown in Figure 2A, the UiO-66-NH_2_ MOFs were synthesized in the presence of MWCNTs and the composites were further used to load a large number of MB molecules. The porous UiO-66-NH_2_ could not only catalyze the oxidation of doxorubicin but also facilitate the adsorption of MB as an internal reference. The ratiometric sensor exhibited higher selectivity and stabilization, in contrast to that with a single-signal response.

Graphene and its analogs have been widely used in the development of electrochemical sensors, owing to its high electrical conductivity, large specific surface area, and excellent chemical stability. A combination of graphene and its analogs with MOFs is an effective approach to increasing the conductivity of MOFs [102,103]. Different hybrid composites of MOFs and graphene analogs (e.g., graphene oxide (GO) aerogel/Cu−MOFs, reduced GO/Cr−MOFs, and graphene/ZIF-67) have been applied to construct sensors for the enzyme-free determination of catechol, 4-nonylphenol, H_2_O_2_, glucose, paraquat, and so on [90,91,104,105,106,107,108]. For example, Wang et al. synthesized the hybrid nanocomposites of copper terephthalate MOF−GO (Cu(tpa)−GO) via an ultrasonication method. The nanocomposites were then used for the determination of drugs (ACOP and DA) [92]. As shown in Figure 2B, the Cu(tpa) was bound with GO through π–π stacking, hydrogen bonding, and Cu–O coordination interactions. After being cast on the glass carbon electrode (GCE), the GO in the composite was electrochemically transformed into reduced graphene with a higher accessible surface area and conductivity. The electroactive Cu(tpa) MOFs in the composites showed a positive influence on enhancing the electrochemical response of ACOP and DA. The LOD for DA was 0.21 μM and that for AP was 0.36 μM. Heteroatom doping can change the electronic structure of GO and improve its electrochemical properties. Recently, Mariyappan et al. developed an electrochemical sensor for chlorogenic acid detection by using Co−MOFs/heteroatoms-doped reduced GO (rGO) [93]. The electron-poor and electron-rich dopants activated the nearby carbon and produced more charged sites for the adsorption of analytes. Moreover, graphene aerogels (GAs) with high porosity and low density offer various sites for binding with MOFs. The combination of MOFs with the GAs matrix can cause a novel synergistic effect for electrochemical sensing [94]. Thus, Lu et al. developed an electrochemical sensor for the simultaneous detection of multiple metal ions in aqueous media by using GAs−supported UiO-66-NH_2_ composites to modify the electrode (Figure 2C) [95]. In this work, GAs enhanced the conductivity and accelerated the electron transfer. Moreover, they acted as the backbones for the in situ growth of UiO-66-NH_2_. The large specific surface area (707.79 m^2^ g^–1^) and the hierarchical porous structure of GA-UiO-66-NH_2_ provided more active sites and mass-transfer pathways for heavy metal ions.

**Figure 2 nanomaterials-12-03248-f002:**
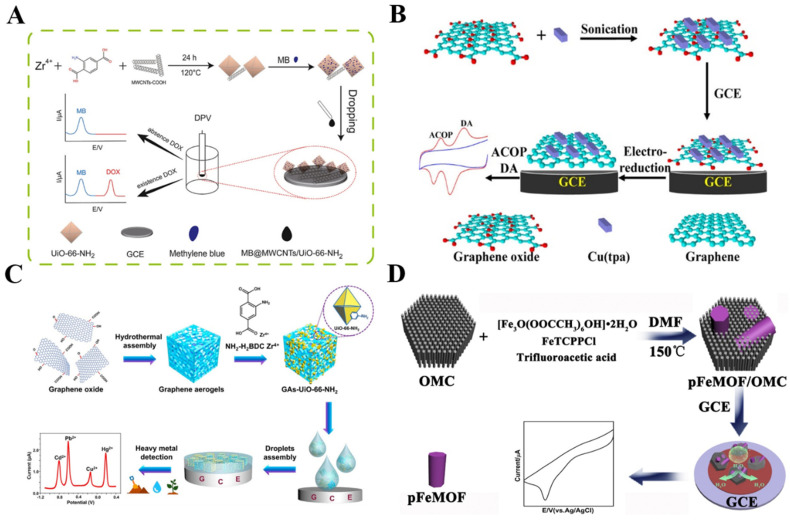
(**A**) Schematic illustration of a ratiometric electrochemical sensor for the determination of doxorubicin, based on MB−loaded MWCNTs/UiO-66-NH_2_. Reprinted with permission from Ref. [89]. Copyright 2022, Elsevier. (**B**) Schematic illustration of the ultrasonication-assisted preparation of Cu(tpa)−GO and the simultaneous determination of ACOP and DA. Reprinted with permission from Ref. [92]. Copyright 2014, American Chemical Society. (**C**) Schematic illustration of the preparation of GAs-UiO-66-NH_2_ and the simultaneous detection of heavy-metal ions by a GAs−UiO-66-NH_2_ modified electrode. Reprinted with permission from Ref. [95]. Copyright 2019, American Chemical Society. (**D**) Schematic illustration of the preparation pathway of pFeMOF/OMC samples. Reprinted with permission from Ref. [99]. Copyright 2017, Elsevier.

Mesoporous carbon (MC), with large pore volumes and superior adsorption capacity, can also be used as a support platform for MOFs. Different MC/MOFs (e.g., MC/Fe−MOFs, MC/Co−MOFs, and MC/ZIF-8) composite-based electrochemical sensors have been reported for the detection of H_2_O_2_, p-nitrotoluene, hydrazine, pyrazinamide, and isonicotinyl hydrazide [97,98,109,110,111,112]. Typically, Deng et al. reported the synthesis in situ of UiO-66 on MC and applied the UiO-66/MC composites to determine the dihydroxybenzene isomers [113]. Zhang et al. demonstrated that the macropores of MC could regulate the homogeneous growth of Co−MOF crystallites, thus providing a large active area for the adsorption of analytes [96]. Ordered MC (OMC), with well-ordered pores, a shorter mesopore channel length, and a large specific surface area can significantly improve the sensitivity of sensors [100,114]. The ZIF-8 and OMC composites were applied to construct an electrochemical sensor for xanthine [101]. In addition, Liu et al. reported the electrochemical detection of H_2_O_2_ in living cells by using porphyrinic iron-based MOF (pFeMOF)/OMC (Figure 2D) [99]. OMC restricted the growth of pFeMOF crystallites and reduced the agglomeration of pFeMOF, providing more active sites when they were exposed to H_2_O_2_. The pFeMOF, self-assembled from Fe^3+^ and porphyrin, possessed a peroxidase-mimic catalytic ability and amplified the electrochemical signal by the reduction of H_2_O_2_.

### 2.3. Noble Metal Nanomaterials

Noble metal nanomaterials, including gold (Au), silver (Ag), and platinum (Pt), have been commonly used as electrocatalysts, to develop various electrochemical sensing devices with high sensitivity and excellent conductivity. The incorporation of precious metal nanomaterials into MOFs can improve the conductivity of MOFs and protect the migration and agglomeration of metal nanomaterials [115]. AuNP−modified MOFs were used to construct an electrochemical sensing platform for the detection of nitrite, methyl mercury species, nitrofurazone, estrone, and DA (Table 3) [116,117,118,119,120]. For example, an AuNP/MOF composite−modified CPE was constructed for the detection of BPA by Silva and co-workers [121]. Mollarasouli et al. prepared porous Cu−MOFs/ZnTe nanorods/AuNPs hybrid composites and used them for the determination of catechol [122]. Wang et al. prepared AuNP−modified MOF (AuNPs/MMPF-6(Fe)) composites via electrostatic interaction and employed them as the electrode materials to detect hydroxylamine with a high electrocatalytic response (Figure 3A) [123]. This work provided a simple method for studying the electrochemical behavior of metalloporphyrin and indicated the potential application of MOF−based composites for bioassays.

In addition, hybrid composites of silver nanoparticles (AgNPs) and MOFs have been used to develop electrochemical assays for the detection of H_2_O_2_, nitrite, peracetic acid, glucose, and so on [125,126,127,128,129]. For example, Peng et al. used MIL-101(Fe) MOFs as the carriers to load AgNPs and then fabricated an electrochemical sensor for Trp detection [130]. The results showed that the combination of MIL-101(Fe) and AgNPs accelerated the electron transfer, thus enhancing the oxidation current of Trp. Liu et al. reported the detection of glutathione (GSH) by using AgNPs to decorate flower-like ultrathin Cu-TCPP MOFs nanosheets [131]. The nanocomposites could increase the electrical conductivity and promote the adsorption of GSH on the sensing interface. Under the synergistic electrocatalysis of MOFs and AgNPs, GSH was sensitively determined in a concentration range that varied from 1.0 μM to 100 μM, with a LOD of 66 nM. Recently, Fan et al. prepared ZIF-67/AgNPs/polydopamine (PDA) nanocomposites with a yolk–shell structure and developed a sensor for the detection of Cl^−^ [132]. In this work, Ag^+^ ions were reduced to AgNPs by DA molecules, and the resulting AgNPs were stabilized by the PDA shell on the ZIF-67/AgNPs surface.

**Table 3 nanomaterials-12-03248-t003:** Detection performances of different metal nanoparticle-modified MOF−based electrochemical sensors.

Electrode Material	Analyte	Linear Range	LOD	Ref.
Pt@UiO-66	H_2_O_2_	5.0 μM~14.75 mM	3.06 μM	[115]
Cu−MOF/Au NPs	NO_3_^−^	0.1 μM~4.0 mM	8.2 nM	[116]
AuNPs@Cu−MOF	BPA	0.2~1.0 mM	37.8 μM	[121]
Cu−MOF/ZnTe NRs/Au NPs	catechol	250 nM~0.3 mM	16 nM	[122]
AuNPs/MMPF-6(Fe)	hydroxylamine	0.01~1.0 μM and 1.0~20 μM	4.0 nM	[123]
AgNPs@ZIF-67	H_2_O_2_	5.0 μM~275 μM	1.5 μM	[126]
AgNPs/MIL-101(Fe)	Trp	1.0~50 μM	0.14 μM	[130]
Ag/Cu−TCPP	GSH	1.0~100 μM	66 nM	[131]
ZIF-67/Ag NPs/PDA	Cl^−^	2.0~400 mM	1.0 mM	[132]
Pt@PMOF(Fe)	H_2_O_2_	0~10 mM	6.0 μM	[124]

**Abbreviation:** AuNPs, gold nanoparticles; BPA, bisphenol A; AgNPs, silver nanoparticles; PDA, polydopamine; Trp, L-tryptophan; GSH, glutathione; PANI, polyaniline.

Besides AuNPs and AgNPs, PtNPs were also decorated on the surface of MOFs as a way to develop electrochemical sensors. For example, Ling et al. reported the preparation of metalloporphyrinic PMOF(Fe) through Fe porphyrin-Zr^4+^ interaction (Figure 3B) [124]. The resulting Pt@PMOF(Fe), with many active Fe centers and dispersed PtNPs on the surface, endowed the nanocomposites with a large surface area and high catalase− and peroxidase−mimicking activities. The PMOF(Fe) acted as a nanocarrier to hinder the aggregation of PtNPs. The Pt@PMOF(Fe) exhibited high catalytic activity for the electrochemical reduction of H_2_O_2_ and O_2_, thus facilitating the design of MOF−based sensing platforms.

### 2.4. Conductive Polymers

The conductive polymers usually involved in polypyrrole (PPy), polythiophene (PTh), and polyaniline (PANI) have been used to modify MOFs for electrochemical assays, due to their simple preparation, good adhesion, high conductivity, and excellent environmental stability [133,134]. Thus, the polymer/MOF (e.g., PANI/Al-MOFs, PPy/Mo-MOFs, PPy/ZIF-8, and PEDOT/MOF-525) composites were prepared and used to detect various targets, such as Zn^2+^, DA, hydroxylamine, quercetin, and so on (Table 4) [135,136,137,138,139]. For instance, Wang et al. developed an electrochemical sensor for the detection of Cd^2+^, based on PANI−functionalized UiO-66-NH_2_ [140]. In the composite, PANI improved the conductivity of MOFs by accelerating the electron transfer. In addition, Xu et al. developed an electrochemical nitrite sensor based on PPy/UiO-66 composites, which were fabricated through in situ oxidative polymerization [141]. The NH_2_-MIL-53(Al) MOFs were electrodeposited on the PPy nanowires and the hybrid nanocomposites were used to modify an electrode for the detection of Pb^2+^ and Cu^2+^ [142]. Due to their good flexibility, electrocatalytic activity, and ease of doping, poly(3,4-ethylenedioxythiophene) (PEDOT) has also been used to modify MOFs [70]. Wang et al. prepared Ni−MOF/PEDOT hybrid composites for the detection of gallic acid and tinidazole [143]. PEDOT acted as the carrier platform for the in situ growth of MOFs and prevented the aggregation of Ni−MOF.

## 3. MOFs as Supporting Platforms

For the fabrication of electrochemical biosensors, biomolecules, such as enzymes, DNA, and antibodies are always used as the recognition elements because of their high selectivity, sensitivity, and signal-to-noise ratio. MOFs with large surface areas, high porosity, and good biocompatibility can be used as excellent porous supports for the immobilization of biomolecules on an electrode surface via covalent or non-covalent interaction, thus improving the stability and reusability of biosensors (Table 5) [144,145,146].

Natural enzymes can catalyze the redox reaction of analytes through an electron transfer between the enzyme and electrode. As a type of electrode modifier for third-generation biosensors, MOFs for enzyme immobilization can not only accelerate the electron transfer but also increase the stability of enzyme molecules during storage and operation [147,148,149,150,151,152,153]. The immobilization strategies can be classified into five groups: physical adsorption, the embedding method, covalent binding, cross-linking, and electrochemical polymerization [154,155]. Various MOFs (e.g., Cu−MOFs, ZIF-8, AuNPs/ZIF-8, MIL-100(Fe)/PtNP, and so on) have been employed to fix glucose oxidase (GOx) and glucose dehydrogenase (GDH) for the detection of glucose [156,157,158,159,160,161]. In the composites, MOFs can act as a secondary biomimetic catalyst for the reduction of produced H_2_O_2_. For example, ZIF-8 MOFs have been employed to prepare enzyme-MOFs composites due to their mild synthesis conditions. Ma et al. used ZIF-8 as the matrix for immobilizing methylene green and GDH for the measurement of glucose [162]. The “single-step” co-precipitation in the biomimetic mineralization method was developed to entrap different enzymes into ZIF-8, including GOx, organophosphate-degrading enzyme A, and a-chymotrypsin [157,158,163]. Recently, GOx and hemin were entrained in a 3D nanocage-based ZIF for the electrochemical detection of glucose via enhanced cascade biocatalysis (Figure 4A) [164]. The outer shell of ZIF prevented the leakage of enzymes, while the interior nanocage provided a second restriction to immobilize the enzymes and maintain their conformational freedom. In this biosensor, GOx catalyzed the oxidation of glucose by O_2_ to generate gluconic acid and H_2_O_2_, which was initiated by the oxidation of hemin. The design of enzyme/MOF−based electrochemical sensors for the detection of H_2_O_2_ is a research hotspot, such as horseradish peroxidase (HRP)/ZIF-67/MWCNTs, HRP/PCN-333(Fe), HRP/ZIF-8/GO, and so on [165,166,167,168]. Gong et al. reported an H_2_O_2_ sensor using a microperoxidase-11/PCN-333(Al) composite [169]. Zhang et al. used the mesoporous and microporous ZIF-8 to immobilize cytochrome c (Cyt c) via electrostatic interaction (Figure 4B) [170]. The adsorption capacity and enzymatic activity of Cyt c were increased when it was immobilized in the ZIF-8 MOFs. An electrode coated with Cyt c/ZIF-8 showed a high sensitivity for the detection of H_2_O_2_ in real samples. Besides this, enzymes/MOFs (e.g., tyrosinase/Ni-Zn-MOFs and tyrosinase/Cu−MOFs) have also been used to construct electrochemical sensing platforms for the detection of pollutants in food and the environment, such as phenol and BPA [171,172,173]. For example, Ma et al. reported the electrochemical determination of BPA, based on multilayer tyrosinase/Cu−TCPP [174].

When the targets are captured by antibodies or aptamers immobilized on the MOFs-modified electrode, the electrochemical impedance would significantly increase, thus inducing a decrease in the voltammetric or amperometric signal, due to the block of mass diffusion of redox probes to the electrode’s surface [175,176,177]. MOFs and their composites (e.g., magnetic Fe_3_O_4_@TMU-21, AuNPs/Zn/Ni−ZIF-8-800@graphene, and Al−MOFs) were used to develop immunosensors for the electrochemical detection of human epidermal growth factor receptor 2, monensin, vomitoxin, and salbutamol [178,179,180]. For example, Biswas et al. reported a label-free electrochemical immunosensor for the detection of carbohydrate antigen 125, based on Zr-trimesic acid MOF (MOF-808)/CNT [181]. The specific antigen-antibody interaction prevented the transfer of [Fe(CN)_6_]^4−/3−^ to the electrode surface. Li et al. reported an electrochemical immunosensor for the detection of alpha-fetoprotein, using AuNPs/ZIF-8 as the support [182].

DNA or RNA aptamers can be immobilized on MOFs via electrostatic interactions, hydrogen bonds, π–π stacking, and van der Waal forces. Many aptamer/MOF−based electrochemical aptasensors have been developed for the determination of zearalenone, lysozyme, cocaine, CEA, and so on [183,184,185,186]. A CuMOF film electrode prepared using an electrodeposition method was used for the dual detection of *Staphylococcus aureus* [187]. As shown in Figure 5A, after the formation of a CuMOF film, AuNPs were electrodeposited on the film to load the DNA aptamers and enhance the electron transfer. When the micrococcal nucleases were secreted by the pathogen, aptamers were cleaved by the nuclease, and the ion movement was accelerated, resulting in an increase in the electrochemical signal of [Fe(CN)_6_]^3−^/^4−^. Meanwhile, the pathogen captured by the aptamer led to a decrease in the electrochemical signal. To avoid the requirement of redox probes in the electrolyte, label-free, and redox probe-free biosensors can be developed by using electroactive MOFs as the substrates [188]. For example, Xing et al. reported a probe-free CuMOF−based electrochemical immunosensor for methyl jasmonate, in which numerous Cu^2+^ ions in the Cu−MOFs produced a strong redox signal directly [189].

Considering that DNA-based homogeneous electrochemical analysis allows target recognition in a homogeneous solution, it is important to develop a continuous homogeneous analysis system for rapid diagnosis and high-throughput bioanalysis. [190]. In homogeneous analysis, the working electrode usually has the ability to capture DNA, but its sensitivity is limited, due to the lack of signal amplification. Thus, Wang et al. developed a homogeneous electrochemical biosensor for the detection of miRNA using Co−MOF nanozyme−modified ITO electrodes (Figure 5B) [191]. The modified electrode could adsorb ssDNA via π−π interactions and repel the MB−labeled hairpin DNA probe. The target miRNA could hybridize with the hairpin (HP) probe to form a DNA/miRNA strand. The DNA probe in the DNA/miRNA strand was cut by duplex-specific nuclease (DSN), then the released miRNA could hybridize with HP again, thus initiating the next cycle. The released MB−labeled short DNA strands were then adsorbed on the surface of MOF/ITO electrode. The MB tags could be electrochemically oxidized by the modified electrode in the presence of H_2_O_2_.

**Table 5 nanomaterials-12-03248-t005:** Detection performances of different electrochemical sensors using MOFs as support platforms.

Type of MOFs	Analyte	Linear Range	LOD	Ref.
Ag@Zn−TSA	H_2_O_2_	0.3 μM~20000 μM	0.08 μM	[148]
ZIF-8@GOx	miRNA-21	0.1 nM~10 μM	29 pM	[149]
Cu−Hemin−GOx	Glucose	10~1555 μM	2.2 μM	[158]
PDA/ZIF-8@rGO	Glucose	1.2 μM~1.2 mM	0.333 μM	[159]
GDH/MG−Tb@MOF−CNTs	Glucose	25 μM~17 mM	8 μM	[161]
MG−ZIFs−GDH	Glucose	0.1~2 mM	Not reported	[162]
GOx/Hemin@NC−ZIF	Glucose	1~20 mM	10 μM	[164]
HRP/ZIF-67(Co)/MWCNT	H_2_O_2_	1.86~1050 μM	0.11 μM	[165]
HRP@PCN-333(Fe)	H_2_O_2_	0.5 μM~1.5 mM	0.09 μM	[166]
ZIFs@HRP/GO	H_2_O_2_	20 μM~6 mM	3.4 μM	[167]
MP-11-PCN-333(Al)−GO	H_2_O_2_	10~800 μM	3 μM	[168]
MP-11-PCN-333(Al)- 3D-KSC	H_2_O_2_	0.387 μM~1.725 mM	0.127 μM	[169]
Cyt *c*@ZIF-8	H_2_O_2_	290 μM~3.6 mM	Not reported	[170]
Tyr@NiZn−MOF NSs	Phenol	0.08 μM~58.2 μM	6.5 nM	[171]
Tyr@Cu−MOFs	BPA	50 nM~3.0 μM	13 nM	[172]
Cu−BTABB−MOF@rGO	BPA	0~100 μM	0.208 μM	[173]
Tyr@Cu–TCPP	BPA	3.5 nM~18.9 μM	1.2 nM	[174]
AuPd NPs@UiO-66-NH_2_/CoSe_2_	Sulfaquinoxaline	1 pg/mL~100 ng/mL	0.547 pg/mL	[175]
Zn−MOF-on-Zr−MOF	PTK7	1 pg/mL~1.0 ng/mL	0.84 pg/mL	[177]
Fe_3_O_4_@TMU-21	HER2	1 pg/mL~100 ng/mL	0.3 pg/mL	[178]
MOF-808/CNTs	CA 125	0.001~0.1 and 0.1~30 ng/mL	0.5 pg/mL	[181]
AuNPs@ZIF-8	AFP	0.1 pg/mL~100 ng/mL	0.033 pg/mL	[182]
MTV polyMOF(Ti)	ZEN	10 fg/mL~10 ng/mL	8.9 fg/mL	[183]
493-MOF	Lysozyme	5 pg/mL~1 ng/mL	3.6 pg/mL	[184]
2D AuNCs@521−MOF	Cocaine	1 pg/mL~1 ng/mL	0.44 pg/mL	[185]
Cu−MOFs	*S. aureus*	7 − 7 × 10^6^ cfu/mL	1.9 cfu/mL	[187]
Cu−MOFs	HBsAg	1~500 ng/mL	730 pg/mL	[188]
Cu−MOF/COOH−GO	Methyl jasmonate	10 pM~100 μM	0.35 pM	[189]

**Abbreviation:** TSA, thiosalicylate; GOx, glucose oxidase; PDA, polydopamine; rGO, reduced graphene oxide; GDH, glucose dehydrogenases; MG, methylene green; CNTs, carbon nanotubes; HRP, horseradish peroxidase; MWCNT, multi-walled carbon nanotube; GO, graphene oxide; MP-11, microperoxidase-11; 3D-KSC, three-dimensional (3D) kenaf stem-derived porous carbon; Cyt c, cytochrome c; Tyr, tyrosinase; BPA, bisphenol A; TCPP, tetrakis(4-carboxyphenyl)porphyrin; PTK7, protein tyrosine kinase-7; HER2, human epidermal growth factor receptor 2; CA 125, carbohydrate antigen 125; AFP, alpha fetoprotein; MTV polyMOF(Ti), multivariate titanium metal–organic framework; ZEN, zearalenone; AuNCs, gold nanoclusters; CEA, carcinoembryonic antigen; S. *aureus*, *Staphylococcus aureus*; HBsAg, hepatitis B surface antigen; COOH-GO, carboxylated graphene oxide.

**Figure 5 nanomaterials-12-03248-f005:**
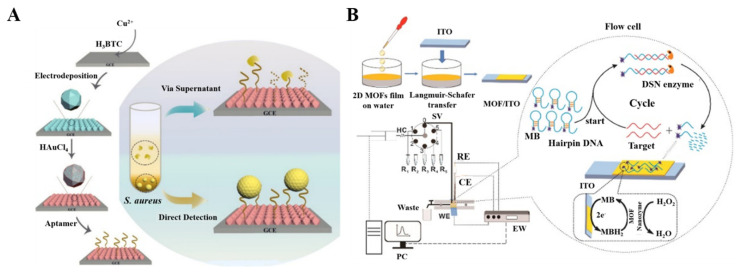
(**A**) Schematic illustration of the fabrication of the DNA/AuNPs/MOFs−based sensing platform and the electrochemical biosensor for the detection of *S. aureus* via supernatants and pathogen cells. Reprinted with permission from Ref. [187]. Copyright 2021, American Chemical Society. (**B**) Schematic illustration of the flow-homogeneous electrochemical assay system for microRNA based on MOF nanozyme. Reprinted with permission from Ref. [191]. Copyright 2022, Elsevier.

## 4. MOFs as Signal Labels

### 4.1. Nanocarriers

Because of their excellent properties, such as a large surface area, high porosity, good adsorption ability, and ease of functionalization, MOFs have been regarded as ideal matrices to load various functional materials for signal output and amplification, including electroactive small molecules, metal ions, enzymes, and nanoparticles (Table 6). Electroactive small molecules (e.g., ferrocene (Fc), 3,3′,5,5′-tetramethylbenzidine (TMB), and MB) are always used as the labels of antibodies or nucleic acids for signal output. However, the weak electrochemical signal and the low marker number may limit the sensitivity of biosensors with electroactive small molecules as the labels. When many small molecules were loaded into MOFs, the signal would be greatly amplified, with the recognition element-labeled MOFs as the signal reporters [192,193]. For this purpose, electroactive molecule-encapsulated MOFs have been used as signal probes for the detection of the p53 gene, N^6^-methyladenine, procalcitonin, exosomes, and so on [194,195,196,197]. For example, the electroactive ferrocenecarboxylic acid was covalently confined in Zn−MOF as the signal tag for the detection of amyloid−β [198]. Li et al. developed an immunosensor for C-reactive protein detection by loading toluidine blue in the channel of the Cu(II)−HKUST-1 [199]. Sun et al. reported an electrochemical biosensor for the detection of glioblastoma (GBM) −derived exosomes, with MB−loaded Zr−based UiO-66 MOFs as the signal reporters [200]. As displayed in Figure 6A, the peptide probes were modified on the electrode for the capture of exosomes by binding to human epidermal growth factor receptor (EGFR) and the EGFR variant (v) III mutation (EGFRvIII), overexpressed on the surface of exosomes. Then, the MB@UiO-66 probes were captured by the electrode, via the interaction between Zr^4+^ ions in the MOFs and phosphate groups on the surface of exosomes. The method could quantify exosomes in the concentration range of 9.5 × 10^3^~1.9 × 10^7^ particles/μL by monitoring the signal change of MB.

The low signal-conversion efficiency based on the ratio of aptamer and target may severely limit the detection sensitivity. Through the DNA−based target cycling amplification, Zhang et al. developed a biosensor for enrofloxacin (ENR) detection by using thionine (Thi)−loaded AuNP-coated bimetallic MOF (Thi−Au@ZnNi−MOF) as the signal label (Figure 6B) [201]. In this study, the signal was amplified by the Pb^2+^-dependent DNAzyme-driven DNA walker. The hairpin DNA (HP1) and the hybrid of wDNA and the ENR aptamer (APT) were self-assembled on an electrode modified with Au&Pt−coated hollow cerium oxide (AuPt@h−CeO_2_) and polyethyleneimine (PEI)−functionalized molybdenum disulfide (PEI−MoS_2_). The APT was released from the electrode surface after binding to ENR. The resulting wDNA could cleave the HP1 in the presence of Pb^2+^, thus resulting in the formation of numerous capture probes. This allowed for the capture of Thi−Au@ZnNi−MOF through an interaction between the capture probe on the electrode and the signal probe on the Thi−Au@ZnNi−MOF. By monitoring the signal change with square wave voltammetry (SWV), ENR was determined with a linear range of 5.0 × 10^−6^ to 1.0 × 10^−2^ ng/mL.

**Figure 6 nanomaterials-12-03248-f006:**
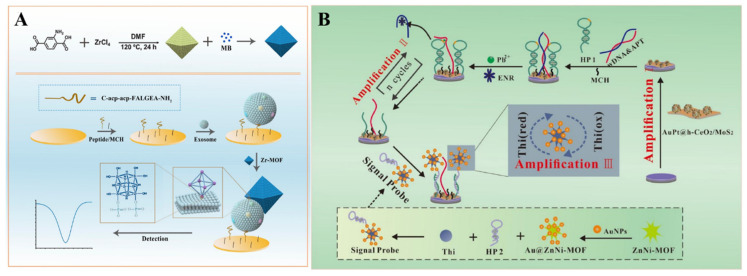
(**A**) Schematic illustration of the fabrication process of eMB@UiO-66-based nanoprobes and the principle of an electrochemical biosensor for the detection of GBM−derived exosomes. Reprinted with permission from Ref. [200]. Copyright 2021, American Chemical Society. (**B**) Schematic illustration of a Au@ZnNi−MOF−labeled electrochemical aptasensor for the detection of enrofloxacin. Reprinted with permission from Ref. [201]. Copyright 2022, Elsevier.

MOFs (e.g., Cu−TCPP, UiO-66-NH_2_) could be employed to carry different signal molecules for the simultaneous detection of multiple targets, such as CEA and carbohydrate antigen 125 (CA125), KANA, and CAP [202,203]. For instance, Chang et al. reported a DNA−functionalized MOF−based homogeneous electrochemical biosensor for the simultaneous determination of multiple tumor biomarkers [204]. As shown in Figure 7A, two electroactive molecules (MB and TMB) were encapsulated into MOFs, and different double-stranded DNA (dsDNA) hybrids were used as the gatekeepers to cap the pore and prevent the release of the electroactive tags, respectively. In the presence of let-7a and miRNA-21, the corresponding dsDNA hybrids were unfolded through the toehold-mediated strand-displacement reaction. Then, the signal molecules were released and quantitatively measured by the electrode. The targets of let-7a and micRNA-21 were simultaneously determined, with an LOD down to 3.6 and 8.2 fM, respectively. Due to the well-defined SWV signals at different potentials, metal ions (e.g., Cd^2+^, Pb^2+^, Zn^2+^ and Cu^2+^) have been entrapped into MOFs as the signal labels for the simultaneous detection of different targets. For example, Yang et al. used UiO-66-NH_2_ to load Cd^2+^ and Pb^2+^ ions for the detection of triazophos and thiacloprid [205]. UIO-66-NH_2_ was used to carry many metal ions (Pb^2+^ and Cd^2+^) as the labels for the simultaneous detection of oxytetracycline and kanamycin [206]. Chen et al. proposed an electrochemical method for the simultaneous determination of multiple antibiotics using an amine-functionalized nanoscale MOF (NMOF) as the signal label (Figure 7B) [207]. In this work, the NMOF was modified with the complementary DNA of the kanamycin (KANA) aptamer (cDNA1) or chloramphenicol (CAP) aptamer (cDNA2). Then, Pb^2+^ and Cd^2+^ ions were loaded on the cDNA1−modified NMOF and cDNA2−modified NMOF, respectively. The two types of NMOFs could be captured by aptamer-modified magnetic beads. In the presence of KANA and CAP, the complementary DNA−modified NMOFs were released and were then determined by SWV under magnetic separation. The detection limits for KANA and CAP were 0.16 pM and 0.19 pM, respectively.

Hemin is a redox mediator with a reversible Fe(III)/Fe(II) redox pair and shows peroxidase-like catalytic activity [208]. However, the dimerization and oxidative decomposition of hemin result in a finite catalytic lifetime. MOFs with high porosity can fix hemin to prevent dimerization and self-destruction. Meanwhile, hemin can endow MOFs with excellent enzyme-mimicking catalytic ability. For example, hemin/Fe-MIL-88 was used as the signal probe for the detection of fibroblast growth factor receptor 3 gene and thrombin [209]. Ling et al. employed Cu−MOFs (HKUST-1) to encapsulate redox-active FeTCPP and the recognition element of streptavidin (SA) for DNA detection (Figure 8) [210]. In the presence of target DNA, the hairpin DNA on the electrode surface was unfolded to form a structure with the combinative SA aptamer. Then, the FeTCPP@MOF−SA composites were immobilized on the electrode surface, based on the specific recognition between SA and its aptamer. The composites could catalyze the oxidation of ophenylenediamine (o-PD) to 2,2′-diaminoazobenzene, producing a high electrochemical signal. This “signal-on” aptasensor showed a wide linear range and an LOD down to 0.48 fM. To avoid the use of the unstable H_2_O_2_ and redox mediate, Xie et al. developed an electrochemical aptasensor for thrombin (TB) detection with hemin-decorated MOFs [211]. Fe−MOFs-NH_2_ was used to encapsulate hemin and AuNPs were used for the conjugation of aptamer and GOx. In the presence of TB, a large number of hemin molecules in Au/hemin@MOFs, captured by the electrode, acted as redox mediators to generate a strong electrochemical signal. The signal could be further amplified by the GOx−catalyzed generation of H_2_O_2_. Finally, the aptasensor exhibited a wide linear range (0.0001~30 nM) and a low LOD (0.068 pM) for TB detection.

MOFs, as enzyme nanocarriers, show high loading capacity. More importantly, MOFs can protect enzymes from biological, thermal, and chemical degradation and still maintain their biological activities. Therefore, enzyme/MOFs, such as GOx/ZIF-8, HRP/Au@Pt/MIL-53(Al), and HRP/Pt/PCN-224 have been used as the signal labels of biosensors to detect CA-242, the COVID-19 nucleocapsid protein, and breast cancer cells [212,213,214]. For instance, Liu et al. developed an electrochemical immunosensor for zearalenone (ZEN) detection, with HRP and IgG-encapsulated ZIFs (HRP/Ab@ZIF-L) as the signal tags (Figure 9A) [215]. The captured HRP/Ab@ZIF-L could catalyze the electrochemical oxidation of TMB in the presence of H_2_O_2_. Li et al. encapsulated GOx and HRP in ZIF-90 for the enzyme cascade reaction to amplify an electrochemical signal [216]. Alkaline phosphatase (ALP) can catalyze the hydrolysis of inert substrates into redox-active products. Recently, Feng et al. designed an electrochemical immunosensor, using Thi− and ALP−loaded ZIFs as the signal tags (Figure 9B) [217]. In this work, bovine serum albumin (BSA) was attached to the surface of ZIF-8 for the immobilization of antibodies and ALP through covalent interactions. ALP catalyzed the hydrolysis of p-aminophosphate ester (p-APP) to produce p-aminophenol (p-AP). The resulting p-AP led to the formation of AgNPs on the electrode surface by the reduction of Ag^+^ ions. Because of the excellent conductivity of AgNPs, an amplified electrochemical signal from the oxidation of Thi was observed. This method allowed for the detection of carbohydrate antigen 72–4 (CA 72–4), with a linear range of 1 μU/mL−10 U/mL.

The hybrid composites of MOFs and nanomaterials can be prepared via encapsulation or physical adsorption, which shows better performance than the single component in electrochemical analysis [218,219,220,221]. For example, Zhong et al. reported the detection of *Escherichia coli* O157:H7 (*E. coli* O157:H7) using CdS QDs−encapsulated ZIF-8 MOFs (CdS@ZIF-8) as the signal probes [222]. As shown in Figure 10A, *E. coli* O157:H7 was captured by the antibody and poly(p-aminobenzoic acid) (PABA)-modified electrode, which allowed for the attachment of the antibody-modified CdS@ZIF-8 probes. Then, a large number of Cd(II) ions were released by treating the electrode with HCl. The released Cd(II) ions were determined by differential pulse voltammetry (DPV). As a result, *E. coli* O157:H7 was detected, with a linear range of 10~10^8^ CFU/mL and an LOD of 3 CFU/mL. However, dissolution by strong acids or oxidants and pre-concentration before electrochemical measurement is complicated and time-consuming. To simplify the experimental procedures, Wang et al. reported the detection of telomerase, using SA−covered AgNP−loaded PCN-224 MOFs (SA−AgNPs/PCN-224) as the signal reporter (Figure 10B) [223]. The elongation of a primer by telomerase led to the formation of an SA aptamer, due to the allosteric activation hairpin probe on the electrode surface. Then, the SA−AgNPs/PCN-224 probe was captured by the electrode through SA−aptamer interaction. A strong electrochemical signal was observed due to the highly characteristic solid-state Ag/AgCl process of AgNPs in the KCl electrolyte medium. The telomerase concentration could be determined by monitoring the signal change from AgNPs.

Metal, bimetallic alloy, and metal oxide nanoparticles are of great significance in electrocatalysis and electrochemical bioassays, due to their excellent electrocatalytic activity and relatively high stability. They have been integrated with MOFs to exhibit higher catalytic efficiency and enhance detection sensitivity, due to the synergistic effect [224]. Electrochemical biosensors based on the composites of MOFs and metal nanoparticles (e.g., hemin−MOFs/PtNPs, AgPt/PCN-223-Fe, and Pd/MIL101-NH_2_) have been developed for the detection of FGFR3 gene mutation, ochratoxin A, telomerase activity, and so on [209,213,225,226,227,228,229]. For instance, Ling et al. synthesized the composites of PtNPs and UiO-66-NH_2_ MOFs via a one-step method and then used them to determine the telomerase activity [230]. Yu et al. developed a Pb^2+^ sensor by using Pd/Pt−modified Fe−MOFs (Fe−MOFs/PdPt) as the signal reporters [231]. As displayed in Figure 11, the substrate DNA strands attached to the rGO−tetraethylene pentamine-gold nanoparticle (rGO-TEPA-Au)-modified electrode were enzymatically cleaved in the presence of Pb^2+^, which led to the formation of short-signal DNA strands. Then, the hairpin DNA-modified Fe−MOFs/PdPt probes were captured by binding them with the signal DNA strands to catalyze the electrochemical reaction of H_2_O_2_.

**Figure 10 nanomaterials-12-03248-f010:**
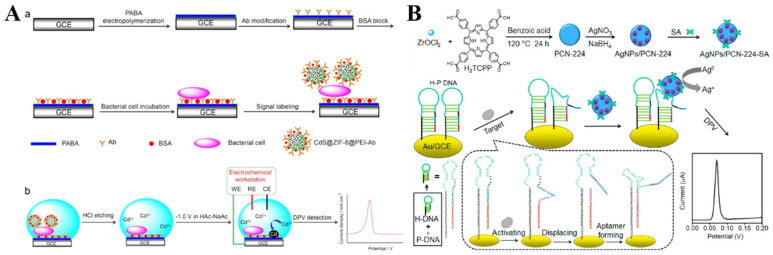
(**A**) Schematic illustration of (a) the fabrication steps of the sandwich−type electrochemical immunobiosensor for the detection of *E. coli* O157:H7 using CdS@ZIF-8 as signal tags. (b) Schematic illustration of the steps for DPV detection. Reprinted with permission from Ref. [222]. Copyright 2019, Elsevier. (**B**) Schematic illustration of the construction of an SA−modified AgNPs/PCN-224 probe and a direct electrochemical sensing strategy of telomerase activity via a conformation switch from aptamer-closed hairpin DNA to aptamer-on hairpin DNA. Reprinted with permission from Ref. [223]. Copyright 2021, Elsevier.

**Table 6 nanomaterials-12-03248-t006:** Detection performances of different electrochemical sensors, using MOFs as nanocarriers.

Type of MOFs	Analyte	Linear Range	LOD	Ref.
Zn−MOF/Fe_3_O_4_–COOH/Thi	CTnI	0.04~50 ng/mL	0.9 pg/mL	[192]
MB@Zr−MOFs	patulin	50 fg/mL~5 μg/mL	14.6 fg/mL	[193]
MB@MIL-101-NH_2_(Cr)	p53 gene	10 fM~100 nM	1.4 fM	[194]
MB@Zr−MOFs	N^6^-methyladenine	1 fM~1 nM	0.89 fM	[195]
UiO-66-TB	procalcitonin	1 pg/mL~100 ng/mL	0.3 pg/mL	[196]
GDH@ZIF-8/[Fe(CN)_6_]^3−^/UiO-66	exosome	1.0 × 10^3^~1.0 × 10^8^ particles/mL	300 particles/mL	[197]
Fc−Zn−MOF	amyloid−β	0.1 pg/mL~100 ng/mL	0.03 pg/mL	[198]
Cu−MOFs−TB	CRP	0.5~200 ng/mL	166.7 pg/mL	[199]
MB@Zr−MOFs	exosome	9.5 × 10^3^~1.9 × 10^7^ particles/μL	7.83 × 10^3^ particles/μL	[200]
Au@ZnNi−MOF	enrofloxacin	5 fg/mL~10 pg/mL	0.102 fg/mL	[201]
Cu−TCPP–TB and PB	CEA and CA125	0.1~160 ng/mL and 0.5~200 U/mL	0.03 ng/mL and 0.05 U/mL	[202]
HP−UIO-66-MB and Fc	KANA and CAP	0.1 pM~50 nM	35 fM and 21 fM	[203]
UiO-66-NH_2_-MB and TMB	let-7a and miRNA-21	0.01~10 and 0.02~10 pM	3.6 fM and 8.2 fM	[204]
UiO-66-NH_2_-Cd^2+^ and Pb^2+^	TRS and THD	0.2~750 ng/mL	0.07 and 0.1 ng/mL	[205]
hemin-MOFs/PtNPs	FGFR3 gene mutation	0.1 fM~1 nM	0.033 fM	[209]
FeTCPP@MOF-SA	DNA	10 fM~10 nM	0.48 fM	[210]
MB−GOx−ZIF-8/Au−rGO	CA 242	0.001~1000 U/mL	69.34 μU/mL	[212]
HRP/hemin/G-quadruplex Au@Pt/MIL-53 (Al)	nucleocapsid protein	0.025~50 ng/mL	8.33 pg/mL	[213]
HRP/hemin/Gquadruplex PtNPs/PCN-224	Cancer cells	20~1×10^7^ cells/mL	6 cells/mL	[214]
HRP/Ab@ZIF-L	ZEN	0.5 ng/L~0.476 μg/L	0.5 ng/L	[215]
GOx/HRP/ZIF-90	CA-125	0.1 pg/L~40 μg/L	0.05 pg/mL	[216]
CdS@ZIF-8	*Escherichia coli* O157:H7	10~10^8^ CFU/mL	3 CFU/mL	[222]
AgNPs/PCN-224	telomerase activity	1 × 10^−7^~1 × 10^−1^ IU/L	5.4 × 10^−8^ IU/L	[223]
Cu_2_O@Cu−MOF@AuNPs	CEA	50 fg/mL~80 ng/mL	17 fg/mL	[224]
AgPt/PCN-223-Fe	ochratoxin A	20 fg/mL~2 ng/mL	14 fg/mL	[225]
Pd/MIL101-NH_2_	telomerase activity	5 × 10^2^~1.62 × 10^7^ HeLa cells/mL	11.25 HeLa cells/mL	[226]
Pd@UiO-66	miRNA-21	20 fM~600 pM	0.713 fM	[227]
PdNPs@Fe−MOFs	miRNA-122	0.01 fM~10 pM	0.003 fM	[228]
Pd@PCN-222	ochratoxin A	10 fg/mL~10 ng/mL	6.79 fg/mL	[229]
Pt@UiO-66-NH_2_	telomerase activity	5 × 10^2^~1 × 10^7^ HeLa cells/mL	2.0 × 10^−11^ IU/L	[230]
Fe−MOFs/PdPt NPs	Pb^2+^	5 pM~1 μM	2 pM	[231]

**Abbreviation:** Thi, thionine; CTnI, cardiac troponin-I; MB, methylene blue; TB, toluidine blue; GDH, glucose dehydrogenase; Fc, ferrocene; CRP, C-reactive protein; TCPP, tetrakis(4-carboxyphenyl)porphyrin; PB, Prussian blue; CEA, carcinoembryonic antigen; CA125, carbohydrate antigen 125; KANA, kanamycin; CAP, chloramphenicol; TMB, 3,3′,5,5′-Tetramethylbenzidine; TRS, triazophos; THD, thiacloprid; SA, streptavidin; GOx, glucose oxidase; rGO, reduced graphene oxide; HRP, horseradish peroxidase; ZEN, zearalenone.

**Figure 11 nanomaterials-12-03248-f011:**
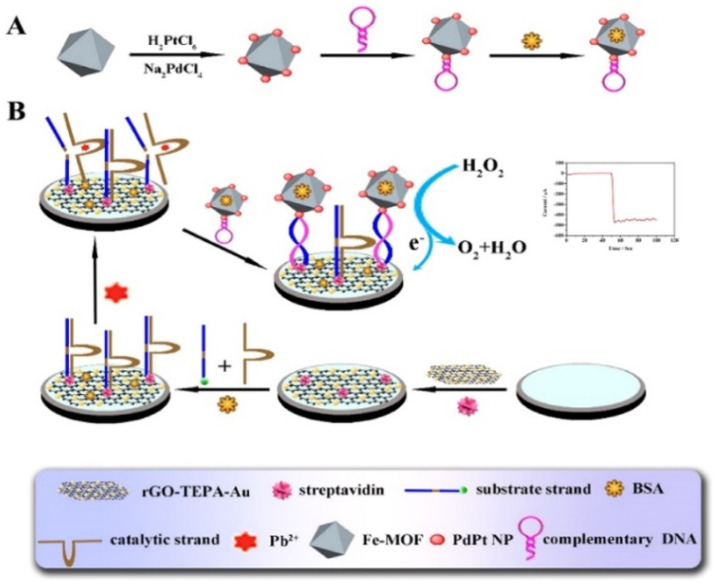
Schematic diagram of: (**A**) the preparation process of Fe−MOFs/PdPt NPs−HP, and (**B**) schematic representation of the proposed strategy for the biosensor. Reprinted with permission from Ref. [231]. Copyright 2018, Elsevier.

### 4.2. Electroactive Labels

The elaborate selection of metal ions and organic ligands can endow MOFs with unique functionality, including optical and electrochemical properties and catalytic activity. MOFs with redox-active metal ions or ligands as the precursors can be directly utilized as the electrochemical signal labels of biosensors without acid dissolution (Table 7). Taking advantage of their specific voltammetric signals at different potentials, MOFs with redox-active metal ions (e.g., Cu−MOFs, Cd−MOFs, and Co−MOFs) are used as the electroactive probes for the detection of endotoxin, CEA, thrombin, prostate-specific antigen, microRNA, and so on [232,233,234,235,236,237,238,239,240]. Liu et al. reported an electrochemical immunosensor for the detection of C-reactive protein by using Cu−MOFs as signal probes [241]. In this study, AuNPs were used to decorate Cu−MOFs with improving conductivity, and PtNP−modified covalent organic frameworks were utilized as the electrode substrates. After the formation of sandwich-like immuno-complexes, the MOFs composites with a large amount of Cu^2+^ ions produced a high electrochemical signal. The immunosensor showed a wide linear range (1~400 ng/mL) for C-reactive protein detection. Recently, Dong et al. reported an aptasensor for Pb^2+^ detection, with PtNPs@Cu−MOF as the signal reporter (Figure 12A) [242]. In this work, DNA walker signal amplification was used to improve the sensitivity through the recognition of an rA site in the DNA walker-substrate strand (SS). In the presence of Pb^2+^, the SS mixture on the electrode surface could be split, thus producing two single S1 and S2 chains. After DNA walker signal amplification, the hairpin DNA−modified PtNPs@Cu−MOF nanocomposites were captured by an electrode covered with the residual DNA fragments. The Pb^2+^ concentration was determined by the current change from Cu−MOF. 

To meet the requirements of the point-of-care (POC) test applications, Chen et al. reported a dual-response biosensor for the electrochemical and glucometer detection of DNA methyltransferase activity, based on invertase−modified Cu−MOFs [243]. As shown in Figure 12B, the Cu−MOFs were sequentially modified with AuNPs, capture probes (CP), and invertase (invertase/CP/Au/CuMOFs). The Au/CuMOFs hybrid composites served as electroactive probes for the production of electrochemical signals and as the support matrixes for the immobilization of invertase. In the presence of Dam MTase, the hairpin probe 1 (HP1) was methylated and then hydrolyzed with the assistance of restriction endonuclease (DpnI). The released binding sequence opened the hairpin probe 2 (HP2) that was immobilized on the electrode surface via hybridization. The exposed sticky terminus on the electrode surface was then hybridized with the CP to tether the invertase/CP/Au/CuMOFs. The electrochemical response of Cu^2+^ ions in Cu−MOFs was recorded by DPV. Invertase in the conjugate catalyzed the hydrolysis of sucrose to generate glucose, which could be readily detected using a personal glucometer.

**Figure 12 nanomaterials-12-03248-f012:**
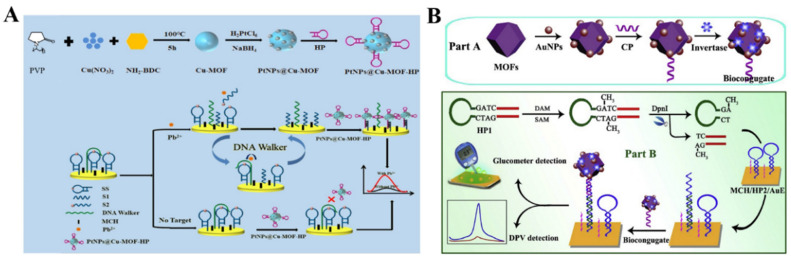
(**A**) Schematic illustration of the fabrication of PtNPs@Cu−MOF−HP/Pb^2+^/MCH/DNA walker-SS/AuE sensor and the mechanism of Pb^2+^ detection. Reprinted with permission from Ref. [242]. Copyright 2022, Elsevier. (**B**) Schematic illustration of the stepwise preparation of invertase/CP/Au/CuMOF bioconjugates and the fabrication process of this proposed dual-response biosensor for Dam MTase activity assay. Reprinted with permission from Ref. [243]. Copyright 2019, Elsevier.

MOFs with NH_2_-BDC as the ligand can be directly used as electroactive signal reporters, which is attributed to the oxidation of amino groups in the ligands [244]. For this view, Li et al. developed an electrochemical biosensor by integration of magnetic separation with nuclease-assisted walking DNA nanomachine [245]. As shown in Figure 13A, the hybrid formed between the MUC1 aptamer and blocker DNA probe (BDP) was immobilized on the surface of a magnetic bead. In the presence of the MUC1 protein, the BDP was released from the surface of the magnetic bead, thus opening the hairpin DNA attached to the AuNPs/MXene−modified electrode. With exonuclease III (Exo III)-assisted cycle amplification, the 3′-end part of DNA in the BDP/DNA hybrid was cut, and the released BDP was moved on the electrode surface to trigger other enzymatic reactions. The residual DNA fragments on the electrode could hybridize with the hairpin DNA probes attached to the AuNPs/UiO-66-NH_2_ MOFs, thus producing an enhanced DPV signal from the oxidation of 2-ATPA ligands in the MOFs. Recently, Dong et al. reported a ratiometric dual-signal electrochemical biosensor for the detection of miRNA, based on Fe−MOFs and UiO-66-NH_2_ [246]. As displayed in Figure 13B, the Fe−MOFs-NH_2_ composites were functionalized, with hairpin H2 probes as the signal tags. The MB-GA-UiO-66-NH_2_ composites were used to modify the electrode for the electrodeposition of AuNPs to immobilize the captured hairpin H1 probes. In the presence of target miRNA-155, the catalytic hairpin assembly (CHA) reaction was triggered, and numerous Fe−MOFs were tethered on the electrode, generating an increased electrochemical signal. Meanwhile, the nanoprobes with poor conductivity hindered the electron transfer, leading to a decrease in the DPV response from MB. The target concentration was determined by monitoring the change in the ratio response of *I*_Fe-MOFs_/*I*_MB_ with a LOD of 50 aM. A ratiometric biosensor with an electroactive species acting as the inner reference probe can improve detection accuracy. Xie et al. reported a sandwich ratiometric electrochemical aptasensor by using Fe−MOF (Fe−MIL-88) and [Fe(CN)_6_]^3−/4−^as the signal reporter and inner reference probe, respectively [247]. A three-dimensional DNA-nanotetrahedron that can eliminate non-specific adsorption from biomolecules was immobilized on the surface of an AuNPs@IL-MoS_2_ electrode for the capture of a target (TB). The signal DNA−modified Au NPs@Fe-MIL-88 could be attached to the electrode surface by binding with the captured target, thus producing an oxidation peak at 0.8 V. With the increase in target concentration, the current from the signal reporter increased, while there was no significant change in the peak current of the inner reference probe. The target concentration was determined according to the ratio response of *I*_signal reporter_/*I*_inner reference_, with a LOD down to 59.6 fM.

**Table 7 nanomaterials-12-03248-t007:** Detection performances of different electrochemical sensors by using MOFs as electroactive labels.

Type of MOFs	Analyte	Linear Range	LOD	Ref.
Cu−BTC MOFs	lipopolysaccharide	1.0 pg/mL~1.0 ng/mL	0.29 pg/mL	[232]
Ag−MOFs	CEA	0.05~120 ng/mL	8 fg/mL	[233]
PtPd NPs/Co−MOFs	thrombin	1 pM~30 nM	0.32 pM	[234]
AuNPs/Cu−MOFs	miRNA-155	1.0 fM~10 nM	0.35 fM	[235]
ZIF-67/ZIF-8	PSA	5 pg/mL~50 ng/mL	0.78 pg/mL	[236]
Cd−MOFs	ochratoxin A	0.05~100 ng/mL	10 pg/mL	[237]
Cd−MOFs-74	p53 gene	0.01~30 pM	6.3 fM	[238]
Cu−MOFs@PtPd NPs	Hg^2+^	0.001~100 nM	0.52 pM	[239]
Co−MOFs@AuNPs	Mucin 1	0.004~400 pM	1.34 fM	[240]
Cu−MOFs@AuNPs	CRP	1~400 ng/mL	0.2 ng/mL	[241]
PtNPs@Cu−MOF	Pb2+	3.0 pM~5 μM	0.2 pM	[242]
Invertase/Cu−MOF	DNA methyltransferase activity	0.002~1 U/mL	0.001 U/mL	[243]
MIL-101(Fe)	telomerase activity	1 × 10^−6^~5 × 10^−2^ IU/L	1.8 × 10^−7^ IU/L	[244]
UiO-66-NH_2_	Mucin 1	5 pg/mL~ 50 ng/mL	0.72 pg/mL	[245]
Fe−MOFs/MB−GA−UiO-66-NH_2_	miRNA	1 fM~100 nM	50 aM	[246]
Fe−MOFs@AuNPs	thrombin	0.298~29.8 pM	59.6 fM	[247]

**Abbreviation:** BTC, 1,3,5-benzenetricarboxylic acid; CEA, carcinoembryonic antigen; AuNP, gold nanoparticles; PSA, prostate-specific antigen; CRP, C-reactive protein; MB, methylene blue; GA, graphene aerogel.

**Figure 13 nanomaterials-12-03248-f013:**
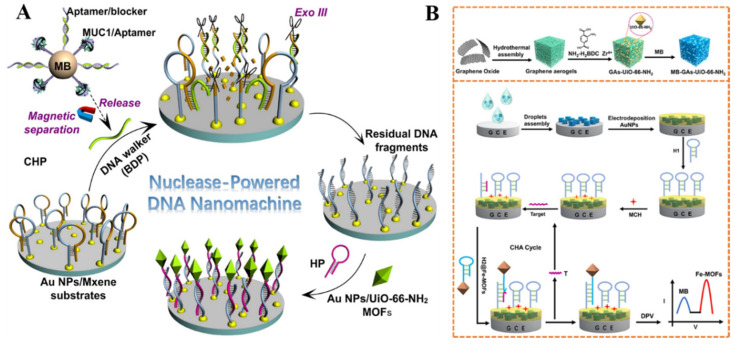
(**A**) Schematic illustration of AuNPs/UiO-66-NH_2_ MOFs-based biosensor for protein biomarkers (MUC1) analysis with the DNA nanomachines powered by Exo III with cycle signal amplification. Reprinted with permission from Ref. [245]. Copyright 2022, Elsevier. (**B**) Schematic illustration of the MB−GA−UiO-66-NH_2_ synthesis procedure and the principle of a ratiometric electrochemical biosensor for the detection of miRNA-155. Reprinted with permission from Ref. [246]. Copyright 2022, American Chemical Society.

### 4.3. Electrocatalysts

Several types of MOFs have been documented to possess enzyme-mimicking catalytic activity, such as Ce−MOFs, Cu−MOFs, and Fe−MOFs, which are denoted as nanozymes. Compared with natural enzymes, these nanozymes have comparable catalytic performances. More importantly, they can remain stable in different, harsh, conditions. Thus, the MOF nanozymes (e.g., Ce−MOFs and Cu−MOFs) have been used to develop electrochemical biosensors for the determination of bacterial lipopolysaccharide, hantavirus, carbohydrate antigen 15–3, and so on (Table 8) [248,249,250,251]. For example, Ce−MOFs can be used as electrocatalysts to design electrochemical biosensors in view of the redox property of Ce^3+^/Ce^4+^. Shen et al. developed an electrochemical aptasensor for lipopolysaccharide detection by using Ce−MOFs and Zn^2+^−dependent DNAzyme-assisted recycling for dual signal amplification [252]. As illustrated in Figure 14A, the CeMOFs were decorated with AuNPs and the thiolated HP2 (HP2/AuNPs/CeMOFs). The presence of lipopolysaccharides triggered the release of the reporter DNA from the duplex. The released reporter DNA as a DNAzyme promoted the circular cleavage of HP1 probes. Then, the HP2/AuNPs/CeMOFs were recruited on the electrode to catalyze the electrochemical oxidation of AA. In addition, the CeMOFs were employed by Yu and co-workers to catalyze the oxidation of Thi for the detection of thrombin [253]. Dong et al. developed a ratiometric CeMOFs-based electrochemical biosensor for the detection of telomerase activity [254]. In this study, the MB−modified hairpin probe was hybridized with telomerase primer (TP) and then anchored to the electrode surface. After the telomerase-catalyzed extension, the hairpin probe was unfolded and the MB tag was farther away from the electrode surface, resulting in a decreased signal. Meanwhile, the CeMOFs/AuNPs catalyzed the oxidation of hydroquinone, leading to a “signal-on” electrochemical response.

Because of their peroxidase-like catalytic ability, porphyrinic MOFs (PCN-222 and (Fe-P)n-MOF) can be utilized as novel electrocatalysts for the analysis of DNA, T4 polynucleotide kinase, and prostate-specific antigen (PSA) [256,257,258,259]. Ling et al. reported a nanoscaled porphyrinic MOF (PorMOF)−based electrochemical biosensor for the detection of telomerase [255]. As shown in Figure 14B, the SA−modified PorMOF was synthesized, with iron porphyrin as the linker and Zr^4+^ ion as the node. In the presence of telomerase, the assistant DNA 1 (aDNA1) was catalytically extended to self-fold into a hairpin structure and the released assistant DNA2 (aDNA2) opened the cDNA through hybridization. Then, the PorMOF@SA nanoprobe was immobilized on the electrode surface through the biotin−SA interaction, which generated an enhanced signal via the electrocatalytic reduction of O_2_. Cui et al. reported the electrochemical detection of Pb^2+^ with PorMOF decorated with AuNPs and Pb^2+^-dependent DNAzyme [260]. In this work, the PorMOF/AuNPs catalyzed the electrochemical oxidation of TMB by H_2_O_2_, greatly amplifying the current intensity.

**Table 8 nanomaterials-12-03248-t008:** Detection performances of different electrochemical sensors by using MOFs as electrocatalysts.

Type of MOFs	Analytes	Linear Ranges	LOD	Ref.
Cu^2+^−NMOFs	lipopolysaccharide	0.0015~750 ng/mL	0.61 pg/mL	[248]
CuMOF	hantavirus	1 fM~1 nM	0.74 fM	[249]
Cu−MOFs@GOx	CA15-3	10 μU/mL~10 mU/mL	5.06 μU/mL	[250]
*p*SC_4_−AuNPs/Cu−MOFs	Fractalkine	10 pg/mL~10 μg/mL	7.4 pg/mL	[251]
AuNPs/Ce−MOFs	lipopolysaccharide	10 fg/mL~100 ng/mL	3.3 fg/mL	[252]
Thi/AuNPs/Ce(III, IV)−MOF	thrombin	0.1 fM~10 nM	0.06 fM	[253]
AuNPs/Ce−MOF	telomerase activity	2 × 10^2^~2 × 10^6^ cells/mL	27 cells/mL	[254]
PorMOF@SA	telomerase activity	1 × 10^2^~1 × 10^7^ cells/mL	30 cells/mL	[255]
PCN-222@SA	DNA	10 fM~100 nM	0.29 fM	[256]
L/(Fe-P)n-MOF	T4 polynucleotide kinase	1.0 mU/mL~1.0 U/mL	0.62 mU/mL	[257]
GR−5/(Fe−P)n−MOF	Pb^2+^	0.05~200 nM	0.034 nM	[260]

**Abbreviation:** GOx, glucose oxidase; CA15-3, carbohydrate antigen 15–3; *p*SC_4_−AuNPs, para-sulfonatocalix [4] arene−coated gold nanoparticles; Thi, thionine; SA, streptavidin.

### 4.4. Sacrificial Templates

Prussian blue (PB), a mixture of ferric and ferrous cyanide, has been widely used in the fields of biomedicine and electrochemistry. It can be used as a redox and electrocatalytic label for bioassays. However, its low stability in water limits the usage of PB in bioassays. Recently, porous Fe−MOFs have been used as metal precursors to produce electroactive PB NPs in situ for signal output and amplification. For example, Bao et al. reported a sensing platform for the detection of miRNA-21, based on the electrochemical conversion of Fe−MOFs into PB NPs [261]. As shown in Figure 15A–C, multipedal polydopamine nanoparticles-DNA (PDANs−DNA) nanomachines were designed that worked on the electrode via multiple legs under exonuclease III-driving. After the formation of DNA dendrimers through the assembly of two hairpins, the Fe-MIL-88-NH_2_ MOFs were immobilized on the electrode. Under a high potential, an acid microenvironment was produced due to the generation of H^+^ on the electrode surface via the water-splitting reaction [262], which made the release of Fe^3+^ ions from MOFs. The released Fe^3+^ ions were then electrochemically transformed into Prussian white (PW) in the presence of Fe(CN)_6_^3−^. The PW was oxidized into PB by O_2_ in the environment. The biosensor could determine miRNA-21 in the concentration range of 10 aM~10 pM with a LOD of 5.8 aM. Based on the conversion of Fe−MOFs into PB NPs in the presence of Fe(CN)_6_^3−^, other biomarkers, such as non-small cell lung cancer ctDNA, T4 polynucleotide kinase, and miRNA-21 have been sensitively detected [263,264,265].

## 5. Conclusions and Future Aspects

In summary, MOF−based nanomaterials have shown outstanding performances in the development of electrochemical sensors. In this review, we comprehensively summarized the different roles of MOFs in the applications of electrochemical sensors. The intrinsic advantages that they offer have promoted the practicability of electrochemical sensors, including large surface area, high porosity, chemical functionality, tunable channel structure, and so on. To overcome the disadvantages of MOF−based nanomaterials (e.g., low conductivity and weak stability), multifunctional conductive materials have been integrated with MOFs to improve the performances of sensors by synergistic effects. Besides, various effective signal amplification strategies have been perfectly integrated into MOF−based electrochemical sensors, such as enzyme catalysis, DNA−based nanomachines, and functional nanomaterials.

Although extensive efforts are being made to improve the performance of MOF−based sensors, certain issues should be further researched to realize the full potential of MOFs. First, the utilization efficiency of electroactive or electrocatalytic metal ions in MOFs should be improved. In recent years, the quasi-MOFs prepared by the controlled deligandation of MOFs and the single-atom catalysts derived from MOFs have attracted extensive attention in different fields. Second, the size of MOFs has an important influence on their activity and stability; however, it is difficult to effectively adjust the size and shape of MOFs, which is unfavorable when comprehensively comparing the performance of MOFs in different projects. Third, due to the complicated structures and compositions of MOFs, the structure-activity relationship has not been clearly investigated and documented. Fourth, the exploration of more redox-active ligands is an urgent need for the development of novel electrocatalysis and electrosensing methods. Lastly, the development of portable devices and simple operation is necessary for in situ and practical applications. Therefore, there are still many undeveloped fields worthy of in-depth and systematic research, in view of the important value of MOFs for the development of electrochemical sensing platforms.

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
