# Peer review of "Design and Application of Electrochemical Sensors with Metal–Organic Frameworks as the Electrode Materials or Signal Tags"

_nanomaterials, 2022, doi:10.3390/nano12183248_

Round 1

Reviewer 1 Report

The review is really interesting, full of schemes. The approach is very clear and comprehensive.

I suggest authors to include:

1) the criteria for choosing the cited articles, which and how many were discarded, which databases were investigated

2) for each paragraph, a final summary table that summarizes the fundamental characteristics of the different sensors

Author Response

We thank the reviewer for his/her positive comments: “The review is really interesting, full of schemes. The approach is very clear and comprehensive.

Comment 1:I suggest authors to include: the criteria for choosing the cited articles, which and how many were discarded, which databases were investigated.

Response: We searched and selected the related articles in Web of Science by using “electrochemical” and “MOF” as the topics. Among them, the related works or highly cited papers were discussed. We have added the following sentences in the revised main text: “Due to the explosive growth of academic articles and the highly dynamic development of this topic, some important papers may be omitted during the above period. Here, we sincerely apologize to the authors whose studies were overlooked in the review.”

Comment 2:For each paragraph, a final summary table that summarizes the fundamental characteristics of the different sensors.

Response: It is a good suggestion. Eight tables have been added to summarize the fundamental characteristics of the different sensors.

Reviewer 2 Report

The paper presents the design and applications of sensors based on MOFs.

It is very interesting research domain for many researchers.

Few minor remarks:

1. Figure resolution should be improved (e.g: fig. 1a, 14a, 15…)

2. The authors should introduce new perspective and trends in the future for sensors based on MOFs

Author Response

We thank the reviewer for his/her positive comments: “The paper presents the design and applications of sensors based on MOFs. It is very interesting research domain for many researchers.

Comment 1:Figure resolution should be improved (e.g: fig. 1a, 14a, 15…)

Response: The re-used figures were copied from the website of article. We have improved the resolution of some figures.

Comment 2:The authors should introduce new perspective and trends in the future for sensors based on MOFs.

Response: We have rewritten the Conclusion to introduce new perspective and trends in the future for sensors based on MOFs.

Reviewer 3 Report

In this work, the author highlighted the recent advancements of MOFs-based electrochemical sensors for the detection of electroactive small molecules and macromolecules. Importantly, the author also highlighted the types and functions of MOFs-based nanomaterials towards electrochemical sensors. To the end , the author provided a conclusion. In overall, this work is potential for publication. I have some minor comments to improve.

1. The author should provide an Scheme to highlight the nature of ligands/building blocks that are used for designing MOFs-based electrochemical sensor.

2. The author should cite very recent review highlights on MOF based sensor Coordination Chemistry Reviews, 2022, 466, 214583.

3. The author should provide a table summarizing all the MOFs and their performance as electrochemical sensor.

4. The author should provide a solid future scope of MOF based electrochemical sensor  with challenges and limitations. It should be a separate section than conclusion

5. A section on general mechanism on the electrochemical sensor using any materials and highlights the importance of MOF specifically towards this application.

Author Response

We thank the reviewer for his/her positive comments: “In this work, the author highlighted the recent advancements of MOFs-based electrochemical sensors for the detection of electroactive small molecules and macromolecules. Importantly, the author also highlighted the types and functions of MOFs-based nanomaterials towards electrochemical sensors. To the end, the author provided a conclusion. In overall, this work is potential for publication. I have some minor comments to improve.

Comment 1:The author should provide an Scheme to highlight the nature of ligands/building blocks that are used for designing MOFs-based electrochemical sensor.

Response: We have added Scheme 1 to highlight the nature of ligands/building blocks that are used for designing MOFs-based electrochemical sensor.

Comment 2:The author should cite very recent review highlights on MOF based sensor Coordination Chemistry Reviews, 2022, 466, 214583.

Response: We have cited the reference.

Comment 3:The author should provide a table summarizing all the MOFs and their performance as electrochemical sensor.

Response: Eight tables have been added to summarize the fundamental characteristics of the different sensors.

Comment 4:The author should provide a solid future scope of MOF based electrochemical sensor with challenges and limitations. It should be a separate section than conclusion.

Response: We have rewritten the conclusion with two paragraphs.

Comment 5:A section on general mechanism on the electrochemical sensor using any materials and highlights the importance of MOF specifically towards this application.

Response: We have revised the Introduction to discuss the general mechanism on the electrochemical sensor using any materials and highlights the importance of MOF specifically towards this application.